# New Lectins from Mediterranean Flora. Activity against HT29 Colon Cancer Cells

**DOI:** 10.3390/ijms20123059

**Published:** 2019-06-22

**Authors:** Isabel Oliveira, António Nunes, Ana Lima, Pedro Borralho, Cecília Rodrigues, Ricardo Boavida Ferreira, Ana Cristina Ribeiro

**Affiliations:** 1Department of Toxicological and Bromatological Sciences (DCTB), Faculty of Pharmacy, Universidade de Lisboa, 1649-003 Lisboa, Portugal; ioliveira2@campus.ul.pt (I.O.); filipe_estesl@yahoo.com (A.N.); 2Linking Landscape, Environment, Agriculture and Food (LEAF), Instituto Superior de Agronomia, Universidade de Lisboa, 1349-017y Lisboa, Portugal; agusmaolima@gmail.com (A.L.); rbferreira@isa.ulisboa.pt (R.B.F.); 3Research Institute for Medicines (iMed.ULisboa), Faculty of Pharmacy, Universidade de Lisboa, 1649-003 Lisboa, Portugal; borralhopm@gmail.com (P.B.); cmprodrigues@ff.ulisboa.pt (C.R.)

**Keywords:** lectins, glycosylated receptors, HT29 colon cancer cells, antitumor activity

## Abstract

Experiments conducted in vitro and in vivo, as well as some preclinical trials for cancer therapeutics, support the antineoplastic properties of lectins. A screening of antitumoral activity on HT29 colon cancer cells, based on polypeptide characterization and specific lectin binding to HT29 cells membrane receptors, was performed in order to assess the bioactivities present in four Mediterranean plant species: *Juniperus oxycedrus* subsp. *oxycedrus*, *Juniperus oxycedrus* subsp. *badia*, *Arbutus unedo* and *Corema album*. Total leaf proteins from each species were evaluated with respect to cell viability and inhibitory activities on HT29 cells (cell migration, matrix metalloproteinase –MMP proteolytic activities). A discussion is presented on a possible mechanism justifying the specific binding of lectins to HT29 cell receptors. All species revealed the presence of proteins with affinity to HT29 cell glycosylated receptors, possibly explaining the differential antitumor activity exhibited by the two most promising species, *Juniperus oxycedrus* subsp. *badia* and *Arbutus unedo*.

## 1. Introduction

Lectins are proteins which are capable of specifically binding to glycans, excluding proteins with catalytic activities, of immunological origin and those involved in transport across membranes. This characteristic makes them unique molecules in cell recognition, especially of anomalous cells carrying on their surface characteristic receptors [1]. In addition, lectins have been described as being capable of inducing cell death in various types of cancer, using several processes such as apoptosis and autophagy. Therefore, since lectins are widely distributed in nature, with plants being one of their largest reservoirs, and exhibit specific properties, they constitute a topic of intense research [2].

Several lectins have been found to possess anticancer properties in vitro, in vivo, and in human case studies. Lectins inhibit the growth of tumor cells by imposing cytotoxic effects (immunomodulatory and anti-metastatic) and mediating apoptosis [3].

Lectins can influence the ultimate fate of various types of cancer cells, culminating in programmed cell death (PCD) via at least three pathways: (a) directly inactivating ribosomes of cancer cells; (b) depending on endocytosis and selectively localizing on some organelles such as mitochondria in cancer cells; (c) binding to specific sugar-containing receptors on the surface of cancer cells. Based on the three possible routes, plant lectins may kill many types of cancer cells via targeting several core PCD pathways including apoptosis and autophagy which, in turn, may provide additional potential new anti-tumor agents for future drug discovery [4]. The ricin B and the GNA-related lectin families act via apoptosis, while Ml-I [4,5] and Con-A [4,6] can act by two different mechanisms: apoptosis and immunomodulation. The lectin BfL [7] and the lectin rich variant mistletoe extract ISCADOR Q [8] exert their effect on matrix metalloproteinase (MMP) -2 and -9 activities.

Several plant lectins such as mistletoe lectins (MLs) and ricin have been well-studied and shown to possess antiproliferative and apoptosis-inducing activities towards cancer cells [9]. In addition, other lectins such as Con A [4,10,11] and PCL [12,13] can result in autophagic cell death after internalization or binding certain sugar-containing receptors on the surface of cancer cells [14].

When lectins act by inhibiting MMP enzymes traditionally associated with matrix remodeling, they can notably affect cell invasion and angiogenesis [15,16]. Previous studies suggest that lectins can indeed exert a strong influence on MMP-9 and MMP-2 [17], although very few have been identified. Therefore, the identification of novel plant lectins with MMPI abilities is of extreme importance.

Colorectal cancer is the third most common cancer in men and the second most common in women. Phenotypic changes may be due to post-translational modifications, namely, changes in protein glycosylation, more specifically, in *N*-glycosylation. *N*-Glycans are of great importance in tumorigenesis, since they are involved in several cellular mechanisms, such as metabolization, signaling, growth, cell adhesion, cell-matrix interaction, invasion and metastasis. Lectins can be used as a tool to decode aberrations on carbohydrates that may be presented by cancer cells. If they exhibit antitumor activity with specificities for *N*-glycans and *O*-glycans, such as T and Tn antigens, Lewis^a^, Lewis^x^ and Forssman antigens [18,19], they could be an excellent means of modulating carcinogenicity.

## 2. Results

### 2.1. Antitumor Activity of Plant Extracts

#### 2.1.1. Cell Viability of HT29 Cells

Total soluble protein extracts from *Juniperus oxycedrus* subsp. *oxycedrus* (Joox), *Juniperus oxycedrus* subsp. *badia* (Joba), *Arbutus unedo* (Aun) and *Corema album* (Cal) were studied in order to evaluate their potential cell death activity on HT29 cells. Figure 1 shows that all species exhibit the capacity to induce cell death and decrease cell viability. However, there are significant differences among species. The *Juniperus oxycedrus* subsp. *oxycedrus* protein extract induced a large increase in LDH activity after 48 h, which was reduced after 72 h, decreasing the MTS metabolism by about 20 to 25% in relation to the control. The *Juniperus oxycedrus* subsp. *badia* protein extract revealed only a very slight effect, since it did not result in a significant difference between the two endpoints in both assays. On the other hand, *Arbutus unedo* and *Corema album* showed very similar results and were the samples with the largest increment in LDH activity; this increase peaked after 72 h. Figure 1b shows a good reduction of the MTS metabolism for these two species; however, *Corema album* was the species with the most pronounced reduction of MTS metabolism after 72 h.

#### 2.1.2. Inhibitory Activities on HT29 Cells

The potential to inhibit the metastatic activity in colon cancer HT29 cells was analyzed by using the migration assay, which evidences the ability that cells have to invade an open gap (wound). In the presence of an efficient inhibitor, the cell gap does not close after 48 h.

Figure 2a shows the pattern of HT29 cell migration after 48 h exposure to protein extracts from Joox, Joba, Aun and Cal and the corresponding representative images of the strongest inhibitions. After 48 h of incubation, an average of 80% migration was achieved in the control, when compared to a much lower proportion of cell migration in the presence of the protein extracts. The highest percentage of cell migration (thus corresponding to the smallest inhibitory effect) was obtained for *Juniperus oxycedrus* subsp. *oxycedrus*, with 68% cutting invasion, as opposed to the lowest percentage of cell migration (thus corresponding to the highest inhibitory effect) recorded for *Arbutus unedo*, with 26%. For the *Juniperus oxycedrus* subsp. *badia* and *Corema album* the percentages of migration were 32.9% and 40.7%, respectively. It may be inferred that there is inhibition of wound closure when protein extracts of the species under study are added to the cell culture medium.

The calculated values presented in Figure 2b are percentage averages obtained after 48 h exposure of the HT29 cells to the different extracts, relative to the first day of exposure, and are indicative of an inhibition of tumor invasion (the smaller the value indicated in Figure 2b, the greater its inhibitory effect on HT29 cell migration), boasting a wound invasion percentage of approximately 30% for the species *Juniperus oxycedrus* subsp. *badia* and *Arbutus unedo*, compared to 80% of wound invasion in the control.

The wound width left open after 48 h was measured for each extract and the mean of the three replicates assayed was calculated. The assays show that the protein extract of *Arbutus unedo* produced the greatest inhibition on cell migration, with 26% shear invasion, which manifests a migration inhibition greater than 50% over the control. The protein extract of *Juniperus oxycedrus* subsp. *badia* invaded the wound by about 33%, promoting a cut-off inhibition of approximately 50%, compared to the control. The species *Juniperus oxycedrus* subsp. *oxycedrus* and *Corema album*, with a cut-off percentage of 68% and 40.7%, respectively, evidenced a lower percentage of inhibition of cell invasion compared to the control.

### 2.2. Proteolytic Activity Inhibition

To explain the inhibition of cell migration, it was decided to investigate the role of gelatinases by having a central role in cell growth. Thus, the work proceeded with all extracts in study, allowing us to evaluate the inhibition of proteolytic activity. In this work, the inhibitory effect of the extracts on MMP-9 activity in the HT29 cell media was analyzed using both gelatin zymography and the DQ-gelatin kit. The results are represented in Figure 3.

The results obtained show that by comparison with the control gel Figure 3a, the gelatinolytic activity is greatly increased in the controls of complete medium (CMC) and physiological saline (CS) of the gel Figure 3b, justifying the free production of MMP-9 and MMP-2 gelatinases by HT29 cells, degrading the gelatin with great evidence. The analysis of the secreted medium after incubation with the extracts under study on gelatin showed a marked inhibition, with evidence of MMP-9 (92 kDa) and MMP-2 (72 kDa) protease bands.

### 2.3. Polypeptide Profile Characterization versus Glycodetection

The protein extracts from leaves of the selected species, *Juniperus oxycedrus* subsp. *oxycedrus* (Joox), *Juniperus oxycedrus* subsp. *badia* (Joba), *Arbutus unedo* (Aun) and *Corema album* (Cal) were characterized by SDS-PAGE in reducing conditions represented in Figure 4. Protein profiles identify specific differences between species, which could be responsible for the differences found in bioactivites.

Through the analysis shown in Figure 4b, it can be deduced that the total protein extracts of *Juniperus oxycedrus* subsp. *badia* and *Arbutus unedo* have glycosylated bands, the first of approximately 50–55 kDa and the second of approximately 45 kDa. The glycosylated band detected in the *Arbutus unedo* extract did not appear in the SDS-PAGE R gel, which shows that the same is not representative, but nevertheless exhibits a strong glycosylation component, making it possible to detect it in the test. The leave extracts of *Juniperus oxycedrus* subsp. *oxycedrus* and *Corema album* have very faint bands, precluding the drawing of conclusions on their representativeness.

### 2.4. Lectin Activity and Sugar Specificity

All plants exhibit a lectin profile as being responsible for diverse bioactivities. The research of lectins is a way of screening extract bioactivities. The presence of lectins is evidenced by haemagglutination assay.

The total soluble protein extracts were prepared and assayed for haemagglutination activity using rabbit and human erythrocytes (Table 1 and Table 2). Generally, 100–150 µg of protein was applied for rabbit erythrocyte assay and 150–250 µg for erythrocytes of human groups (A, B and 0). Haemagglutination activities v.s. inhibition of haemagglutination were analyzed (Table 1 and Table 2), in order to give an identity to the lectins in the protein profile.

Analyzing Table 1, it may be concluded that only *Juniperus oxycedrus* subsp. *oxycedrus* and *Juniperus oxycedrus* subsp. *badia* show specificity for the human blood groups, ARH+ and ARH-, with a H.U. of 16.6 µg and 250 µg respectively for Joox and 50 µg and 27.7 µg for Joba.

The haemagglutination activity for *Juniperus oxycedrus* subsp. *badia*, *Arbutus unedo* and *Corema album* total protein extract on rabbit erythrocytes show that all species have erythrocyte haemagglutination which means the presence of lectins, with Aun and Cal as the species with highest levels of activity.

The standard assay for haemagglutination inhibition on rabbit erythrocytes was performed for 3 species for a panel of 17 sugars at 0.1 M concentration, except fucose (0.3 M), with 4 H.U. of protein. The results are presented in Table 2.

The carbohydrate specificity, exhibited by *Juniperus oxycedrus* subsp. *badia*, identified by haemagglutination inhibition assays on rabbit erythrocytes, with 4 H.U. (0.820 µg), by decreasing order was: in ex aequo D-mannose, α -methyl-D-glucopyranoside, melezitose (all with m.i.c. = 5.6 × 10^−7^ M), α-methyl-D-mannopyranoside (m.i.c. = 5.1 × 10^–7^ M), sucrose (m.i.c = 4.6 × 10^–6^ M), L-(-)fucose (m.i.c. = 0.4 × 10^–4^ M), galactosamine and glucosamine (both with m.i.c. = 3.7 × 10^–3^ M), L-(+)fucose (m.i.c. = 0.1 × 10^–3^ M) and *N*-acetyl-D-glucosamine (m.i.c. = 11.1 × 10^–3^ M).

For *Arbutus unedo* the species with more lectin activity, the assay for 4 H.U. (0.180 µg) revealed carbohydrate specificity by decreasing order: melezitose and α-methyl-D-mannopyranoside (both with m.i.c. = 2.74 × 10^–3^ M), D-glucosamine (m.i.c. = 8.2 × 10^–3^ M), D-glucose, D-mannose and maltose (all with m.i.c. = 0.074 M), *N*-acetyl-D-glucosamine, D-galactosamine and α-methyl-D-glucopyranoside (all with m.i.c. = 0.025 M), L(-)-fucose (m.i.c. =0.011 M) and sucrose (m.i.c. = 0.1 M).

*Corema album* is the species with fewest specific sugars that cause haemagglutination inhibition. For 4 H.U. (0.276 µg) assayed, the carbohydrate specificity by decreasing order was: α-methyl-D-mannopyranoside and melezitose (both with m.i.c. = 1.5 × 10^−5^ M), D-mannose (m.i.c. = 3.7 × 10^–3^ M), α-methyl-D-glucopyranoside (m.i.c. = 0.41 × 10^–3^ M), sucrose (m.i.c. = 0.011 M) and D-glucose, *N*-acetyl-D-glucosamine and D-galactosamine (all with m.i.c. = 0.1 M).

### 2.5. Lectin Specificity for HT29 Cell Membrane Glycosylated Receptors

The affinity binding of lectins from different extracts to HT29 cells was validated by 2D electrophoresis (IEF/SDS-PAGE).

A comparative analysis was performed between 2D electrophoresis of HT29 cell membrane control and 2D electrophoresis of membrane incubation with leaf extracts of *Juniperus oxycedrus* subsp. *oxycedrus*, *Juniperus oxycedrus* subsp. *badia*, *Arbutus unedo* and *Corema album*. The differences between the peptide spots recorded in Figure 5a control and each panel b, c, d and e reveal new peptide spots, meaning that lectin-like proteins present on of the different extracts bind to the cell receptors.

*Juniperus oxycedrus* subsp. *oxycedrus* extract exhibits 3 peptide spots that established a specific binding to HT29 glycosylated receptors (Figure 5b). Two of the spots have a pI of approximately 6.0 with a molecular mass around 30 kDa and 15 kDa. The remaining spot comprises a pI close to 7.0, having a molecular mass of approximately 32 kDa.

*Juniperus oxycedrus* subsp. *badia* extract evidenced 3 peptide spots (Figure 5c). The spots with pIs close to 6.0 reveal the same molecular mass of the spots identified in Figure 5c, and may indicate a characteristic spot, like a molecular marker, of the proteomic profile of *Juniperus oxycedrus* species. The third spot comprises a molecular mass of approximately 50 kDa, with a pI close to 7.0.

*Arbutus unedo* extract reveal 3 peptide spots (Figure 5c). Two of the spots have similar molecular weights, i.e., of between 44 to 46 kDa and a pI of 5. The third spot identified has a pI of 8.0 and a molecular mass of approximately 15 kDa.

*Corema album* extract showed that there were at least two bands that bound to the membranes of HT29 cells. These spots have an acidic pI of between 4 and 5 and a molecular mass of 13 kDa and 90 kDa, respectively (Figure 5e).

## 3. Discussion

In recent decades, we have witnessed growth in the importance of screening plant compounds for new anticancer activities. Natural product research complemented with ethnobotanical and ethnopharmacological knowledge of the Mediterranean flora can be of potential use to identify novel candidates with desirable bioactivities to develop new cancer therapies.

All cellular processes involve proteins, and therefore, the identification of proteins and the characterization of their functional activities become increasingly important to understand the biological processes of living beings. The studies performed on ethnobotanical species have neglected, to some extent, protein bioactivities and, more particularly, lectin bioactivities.

This work involves a screening of potential antitumor activity towards colon cancer cells of different leaf protein extracts from endemic Mediterranean flora species, namely *Juniperus oxycedrus* subsp. *oxycedrus*, *Juniperus oxycedrus* subsp. *badia*, *Arbutus unedo* and *Corema album*. For each species, the cytotoxic activity on HT29 colon cancer cells and the MMP inhibition were evaluated, leading to the hypothesis of a possible mechanism mediated by lectins of cellular death.

To assess the bioactivities of the different leaf protein extracts, HT29 cell viability, inhibition of tumor invasion, and MMP-9 and MMP-2 inhibitory activities were assessed.

In the cell viability assays, the *Arbutus unedo* and *Corema album* total protein extracts were the two most promising candidates, since they showed a better capacity to induce cell death and decrease cell viability. The inhibitory activity results showed that *Juniperus oxycedrus* subsp. *badia* was the most effective at inhibiting cellular migration, and this extract also exhibited a promising inhibition of MMP activity as compared to the control. *Arbutus unedo* also showed good results in these two assays. These experiments have not only shown that all species under study display antitumor activity on HT29 cells, but more importantly, they highlight the marked differences among species.

A screening on MMP-9 inhibitory activity by the protein leaf extracts of *Juniperus* species, *J. oxycedrus L*. subsp. *oxycedrus*, and *J. oxycedrus* subsp. *badia*, showed that the proteolytic inhibition ability significantly differed (*p* < 0.001) between species and between extracts as well. Firstly, *J. oxycedrus oxycedrus* did not influence MMP-9 activity whilst the other species did, at different levels. The inhibitory activities achieved for *Arbutus unedo* and *Corema album* extracts were similar (around 50%) but lower than that observed for *J. oxycedrus* subsp. *badia*, which yielded a reduction of 95% on MMP-9 activity. Again, these results not only show that *Juniperus* species can inhibit MMP activity, but more importantly, they highlight the marked differences among species. For the protein extracts, results show that only *J. oxycedrus badia* (Joba) was unable to dramatically reduce MMP-9 activities. Considering that MMP comprise an enzyme family deeply associated to cancer metastasis and progression, and also to pre-cancer inflammatory states, and since our aim was to identify lectin-based MMPIs, the work was pursued with all species using HT29 colon cancer cells. By comparing all species, we hoped to pinpoint specific differences among them that may provide clues for the high activities found in Joba.

The antitumor activity data can be explained by the inhibition of cell migration in several ways: MMP-9/MMP-2 metalloproteinase production inhibition or inactivation caused by the protein extracts under study; growth factor inhibition; and/or the presence of compounds that elicit the release of mediators aimed at preventing cell adhesion and invasion as well as programed cell death mechanisms. The glycan component of MMPs may constitute a substrate for lectin activity, allowing formation to occur of a specific bond that can contribute to a mechanism of antitumor action.

After determining the inhibitory activities on HT29 colon cancer cells by all species under study, the potential to inhibit the metastatic activity of these cells was analyzed using the migration assay, which shows the ability that cells have to invade an open gap (wound). Unlike the control, in the presence of an efficient inhibitor, the cell gap does not close after 48 h. In Figure 2 we can observe the pattern of HT29 cell migration after 48 h exposure to protein extracts from Joba, Joox, Aun and Cal, and the respective representative images of the highest inhibitions.

Previous reports have already determined the effects of crude aqueous extracts of a panel of medicinal plants on the growth and invasion of other types of cancer cells and showed that *Juniperus communis* significantly decreased the growth of MCF-7/AZ breast cancer cells [20]. Other experiments also showed the cytotoxic effects of several *Juniperus* species phenolic compounds on different cancer cell lines [21]. Our results corroborate that cell invasion is affected by the presence of *Arbutus unedo* and *Juniperus oxycedrus* subsp. *badia* protein extracts, and to a far greater extent (*p* < 0.05) of Joba extract. Recent studies showed that phenolic compounds from *Juniperus* could induce apoptosis in HT29 cells [22,23]. In this work, no cytotoxic effects were detected with protein extracts, but the significant reduction in cancer cell invasion and the results obtained in Figure 3 suggest that these results could be related to MMP-9 inhibition abilities observed in Joba. Hence, the inhibitory effect of the extracts on MMP-9 activity in the HT29 cell media was also analyzed using both the DQ-gelatin assay and gelatin zymography.

Gelatinolytic activities (Figure 3c) indicate that gelatinases from HT29 cells were strongly inhibited by protein extracts from Joba, but not from the other plant extracts. Furthermore, zymographic analysis of the cell media exposed to Joba extracts (Figure 3b) further demonstrated that expression of both gelatinases (i.e. MMP-9 and MMP-2) in HT29 cells was inhibited when compared to controls. The difference found in both *Juniperus* subspecies substantiates once more a higher activity in Joba, which might be specific to this subspecies. In order to identify the proteins which could be responsible for this difference between subspecies, and knowing that plants are the largest known natural reservoir of lectins, the study was continued with all species in order to identify lectin-like proteins in the different extracts, in an attempt to study their potential role in the observed differential bioactivities. The polypeptide pattern was assessed, as well as the glycan profile. Since the difference in protein profiles could be responsible to the difference found in bioactivities, the lectin-like activities of all extracts were determined, using haemagglutination assays and HT29 cell membrane specific binding assays.

In lectin activities and sugars specificity, the haemagglutination activity for all plant extracts were used for quantify total lectin activity. Two kinds of erythrocyte cells were tested. The haemagglutination of human erythrocytes for all species revealed that only two species recognize the glycosylated receptors on erythrocyte cell membranes, implying one other difference among the species under study. Only Joox and Joba allowed haemagglutination activity with ARh+ and ARh- groups (Table 1). For the inhibitory studies of haemagglutination by sugars, none of the sugars assayed inhibited the haemagglutination activity, meaning that the lectins present did not show specificity for those sugars at the concentrations applied (4 H.U.).

The human blood groups are distinguished by their glycosylated receptors on the cell surface bound to glycolipids or glycoproteins. The human group 0 has L-fucose as the terminal glycosylated receptor, the group B expresses galactose, and group A has GalNAc alpha I receptors [24]. As group A contain GalNAc residues, it was expected that this sugar, as well as sugars like galactose and sugars with alpha-galactosyl residues (raffinose) could be haemagglutination inhibitors for the activity of these lectins. The assays of inhibition of haemagglutination on human erythrocytes did not reveal the presence of lectins with specificity for any kind of sugars assayed, thus exhibiting haemagglutination in the presence of all sugars. This observation may be tentatively explained if we assume that a low sugar concentration (0.1 M) was used in haemagglutination activity assays.

For rabbit erythrocytes, the haemagglutination activity detected for a panel of 17 sugars at 0.1 M concentration with 4 H.U. of protein revealed that Joba extracts exhibited a strong specificity towards D-mannose (m.i.c. = 5.6 × 10^−7^ M) and one related sugar containing mannose, α-methyl-D-mannopyranoside (m.i.c. = 5.1 × 10^−7^ M). Carbohydrates with glucose in its composition, such as melezitose and α-methyl-D-glucopyranoside, (both with m.i.c. = 5.6 × 10^–7^ M) and sucrose (m.i.c. = 4.6 × 10^−6^ M) have equally a hight specificity, while L-fucose and *N*-acetyl-D-glucosamine, sugars with intervention in tumor processes, exhibit less specificity (with m.i.c. = 0.1 × 10^–3^ M and m.i.c. = 11.1 × 10^−3^ M, respectively). The sugar raffinose, a trisaccharide with galactose in its composition, showed a minimal inhibitory concentration of 0.4 × 10^−3^ M. The sugars D-galactosamine, D-glucosamine and D-glucose exhibited a weaker inhibitory power, whereas other sugars like galactose, *N*-acetyl-D-galactosamine, lactose, maltose and sialic acid did not reveal any inhibitory effect (Table 2). The *Arbutus unedo* and *Corema album* also presented inhibitory capacity of haemmaglutination, but with lower m.i.c. values for D-mannose, melezitose, α-methyl-D-mannopyranoside and α-methyl-D-glucopyranoside.

The results for haemmaglutination and the inhibition of haemagglutination activities in the protein extracts revealed the presence of lectins in the four plants under study, probably different lectins with different specificities to carbohydrates.

When interpreting our data, we must consider that lectins in general tend to have a strong specificity to bind complex carbohydrates when compared to monosaccharides, often justifying the low values of haemmaglutination inhibition detected with monosaccharides [1]. In the present assays, the selection of the sugar panel was made with two main objectives: (i) to try to classify the lectins present in our samples based on their sugar specificity; and (ii) to perceive their specificity for characteristic sugars present on glycomic aberrations of tumor cells, namely branched *N*-glycans, *O*-glycans and fucose.

Our results suggest that we have detected four types of lectins in our protein extracts since, according to the sugar panel inhibition tests, we detected inhibition for group IV (L-fucose); the inhibition observed for glucosamine and galactosamine could mean that core molecules, glucose and galactose, can be part of the other groups like mannose/glucose (groupI), galactose/*N*-acetyl-galactosamine (group II) and *N*-acetyl-glucosamine (group III), from the lectins classification by sugar specificity. This highlights the potential of lectins in the recognition of tumor cells, since these glycans are important in tumor development.

In aberrant glycosylation several sugars assume a leading role, as β1, 6 branched and bisected *N*-glycan that modulates cell behavior by interfering in the physical and performance properties of glycosylated adhesion molecules on the cell surface; a poly-*N*-acetyl-lactosamine containing glycans, that can be potentially recognized by galectins; and an outer-chain polyfucosylation and sialyl Lewis^x^ production that are potentially recognized by the selectins [18].

Some lectins as SNA (*Sambucus nigra* agglutinin), VFA (*Vicia faba* agglutinin), WGA (wheat germ agglutinin) [25] and ACA (*Amaranthus caudatus* agglutinin) [26] have been described as having antitumor activity on HT29 colon cancer cells, as cytotoxicity, aggregation, binding to cell membrane glycosylated receptors, adhesion and tumor inhibition [4,10]. The glycan specificity of some of these lectins is similar to the specificity detected for protein extracts of Joba, providing a good perspective for its antitumor activity on HT29 cells.

Other lectins, with similar carbohydrate specificities to the lectins detected but without detected bioactivity on HT29 cells, also have an effect on others tumor cells lines. In cancer process, the glycomic aberration at the level of the cell surface carbohydrates is one of the largest changes suffered by cancer cell [27]. The overexpression of branched *N*-linked β-1,6-GlcNAc oligosaccharides (due to the increased activity of GlNAc-TV) [28] that increases the total cell surface terminal sialylation in malignant cells [29,30], as many other carbohydrates structures do, such as sialylLewis^a^ (sLe^a^) and sialylLewis^x^ (sLe^x^), sialylated fucosylated structures [31], present in *O*-linked and *N*-linked oligosaccharides, have been shown to be displayed in various human malignancies [30,32], with associations with advanced forms of malignancies and poor prognoses in breast, bladder, lung and colon carcinomas. Recently, Nakajima and colleges [33] appointed a new predictive biomarker for distant recurrence of curatively resected colorectal cancer based on a carbohydrate receptor detected by ABA lectin (Agaricus bisporus agglutinin) whose specificity is for Galβ1-3GalNAc and *N*-acetylgalactosamine. Some carbohydrate specificity found by lectins present on three of the tested extracts suggests that L-fucose and N-acethylglucosamine are usually aberrant forms of carbohydrates present in surface cell membrane of HT29 cells, and therefore, they will give a bad prognostic in tumoral process. Nevertheless Con-A, a lectin with mannose/glucose specificity (one of the glycan moieties for which *Juniperus oxycedrus* subsp. *badia* extracts exhibit a strong specificity, has an antitumor effect exerted by autophagy and apoptosis.

The evaluation of the specific binding of peptides/polypeptides to HT29 cells membranes was made by 2D proteomic analysis as a way of revealing the lectins involved in binding to glycosylated receptors (Figure 5). For *Juniperus oxycedrus* subsp *oxycedrus* three polypeptide spots were noted, indicating possible links of lectin type proteins to glycosylated membrane receptors from HT29 cells. All of the spots (15, 30 and 32 kDa) showed a pI around neutrality. Analyzing the polypeptide profiles obtained under reducing conditions (Figure 4b, lane 1), we noted the presence of these polypeptide bands, indicating the matching of these polypeptides with the protein extracts.

For the extract of *Juniperus oxycedrus* subsp. *badia*, also three polypeptide spots (15, 30 and 50 kDa) were detected which were bound to the glycosylated membrane receptors from HT29 cells. The bound polypeptides also have a pI around neutrality. The spots with pIs around 6.0 revealed the same molecular masses (15 and 30 kDa) as the spots identified in Figure 5b. This observation may indicate a feature of the proteomic profile of the species *Juniperus oxycedrus* that could be considered a molecular marker, as mentioned. The third spot marked comprises a polypeptide with a molecular mass of approximately 50 kDa, with a pI around 7.0, being a glycoprotein (results not shown). All spots have correspondence with the electrophoretic profile obtained under reducing conditions (Figure 4b, lane 2). The presence in Joba extract of a specific polypeptide spot of 50 kDa that binds to the HT29 cell membrane assumes a particular interest, since it was unique in Joba and exerted a possible lectin bioactivity.

For Aun extract (Figure 5d), three new spots were identified with 15, 44 and 46 kDa, and pI 8.0 and pI 5.0 for the last two. Cal 2D analysis (Figure 5e) revealed binding of two peptide spots of 13 and 90 kDa and pI 4.0 and pI 5.0, respectively.

The evaluation of the lectin character of the specific peptides/polypeptides bound to HT29 cells membrane proteins from all protein extracts is supported by haemagglutination and inhibition by sugars, exhibiting the modus operandi of lectins, established by affinity binding to specific glycans of the cellular glycoma. The specific glycans identified for each species included glycans that are present in cancer aberrations, making these lectin-like proteins promising partners in the detection and decoding of this aberration.

To date, there are no studies that involve lectins from the species under study. The binding of lectins to different cancer cell lines and exerting an antitumor mechanism of action are already well documented in other species. The preferred target of lectins are glycosylated receptors present on the surface of the cell membrane, usually in the form of glycoproteins or glycolipids. The relative proportions of these glycoconjugates are distinct on different membranes, reflecting the diversity of their biological roles. Qualitative and quantitative changes in glycoproteic components of cell membranes are highly significant in the development and progression of many neoplastic processes [26].

Studies of cell surface carbohydrates from colon cancer tissues showed modifications of *N*-bound glycans at β-1,6-GlcNAc branched oligosaccharides at the beginning of the tumor process by oncogenes [32,34]. Often, the terminal of the β-1,6-GlcNAc branched oligosaccharides are fucosylated or sialylated in malignant cells. The inhibition of the haemagglutination assays on rabbit erythrocytes showed specificity to *N*-acetyl-D-glucosamine and L-fucose, which could explain the binding of these polypeptides as lectins, from the protein extracts, to the glycosylated receptors on HT29 membranes, meaning that these lectins could be good tools to exert some action on HT29 colon cancer cells.

Hence, the specific binding of the polypeptides from *J. oxycedrus badia* and *Arbutus unedo* extracts to the membranes of HT29 colon cancer cells, associated with the initial results on cell viability, cell invasion in the migration assays, and the presence of lectins in extracts, may be good indicators of the inhibitory action of these proteins on HT29 cancer cells.

## 4. Materials and Methods

### 4.1. Plant Material and Animal Cells

#### 4.1.1. Plants

*Juniperus oxycedrus* subsp. *oxycedrus*, *Juniperus oxycedrus* subsp. *Badia and Arbutus unedo* species, voucher samples were authenticated and deposited at the herbarium João de Carvalho e Vasconcelos, Instituto Superior de Agronomia, Lisbon, Portugal. All plants were grown in pots, exposed to the same local environmental conditions of light, temperature, and humidity. Plants were watered when necessary. The *Corema album* leaves were harvested in Comporta, in October 2010. All leaf species samples were collected and stored at −80 °C for later analysis.

#### 4.1.2. HT29 Cells

The HT29 colon adenocarcinoma from *Homo sapiens sapiens*, cell line ECACC, No. 91072201, were used in in vitro experiments.

#### 4.1.3. Erythrocyte Cells

Rabbit erythrocytes were commercially available (*Probiológica*), and human erythrocytes (group A RH+, A RH-, B RH- and 0 RH-) were collected from healthy donors with consent by a technical expert. Both blood cells were used for the hemagglutination assays.

### 4.2. Leaf Protein Extraction

The total protein extract of leaves from *Juniperus oxycedrus* subsp. *oxycedrus*, *Juniperus oxycedrus* subsp. *badia* and *Corema album* were obtained by Jacobs method [35]. The leaves were powdered under liquid nitrogen containing polyvinylpolypyrrolidone (0.5 g PVPP per g of fresh weight), were add 10 mL·per 1 g fresh weight, at 4 °C, of Tris-HCl 350 mM, pH 8.0, containing 11 mM sodium diethylcarbamate and 15 mM cysteine. A protease inhibitor cocktail for complete inhibition of serine and cysteine proteases (Complete, EDTA free inhibitor from Roche Switzerland) was added at the beginning of the extraction procedure (1 tablet of cocktail inhibitor for each 10 mL of extraction buffer). The slurry was centrifuged at 18,000× *g* for 15 min at 4 °C (Beckman J2–21M/E, rotor JA 20000), and the supernatant desalted through PD-10 columns (GE Healthcare), with a Sephadex G-25 Medium previously equilibrated in 50 mM Tris–HCl buffer (pH 7.5) containing 2 mM CaCl_2_ and 2 mM MgCl_2_, and dialyzed against saline containing 2 mM CaCl_2_ and 2 mM MgCl_2_.

The total protein extract from *Arbutus unedo* L. were obtained by an optimized method of extraction with buffer containing 5% glycerol and 1% Triton X-100, described by Silva and Souza [36].

### 4.3. Antitumoral Activity

#### 4.3.1. HT29 Cell Culture

The HT29 cells were routinely seeded in plastic flasks and cultured in culture media Roswell Park Memorial Institute (RPMI), containing 2 mM Glutamine, and supplemented with 10% (*v*/*v*) fetal calf serum, 0.5% (*v*/*v*) of Penicillin solution at 2 × 10^4^ UI/mL and 34 mM Streptomicin. The cells were kept in a humidified atmosphere containing 5% CO_2_. The cells were supplied with fresh medium every second day, and trypsinized when the confluence was between 80–100%.

#### 4.3.2. Measurement of Cell Death and Viability by MTS and LDH

The evaluations of cell viability and cell death were performed using the MTS metabolism and lactate dehydrogenase (LDH) assays, respectively. These assays were performed in parallel. 5 × 10^3^ cells per well, in 96-well plate, were incubated with 100 µg/mL of each total proteins extracts of the species under investigation and with saline containing 2 mM CaCl_2_ and 2 mM MgCl_2_ as a control, for 48 and 72 h.

General cell death was evaluated using LDH Cytotoxicity Detection KitPLUS (Roche Diagnostics GmbH, Germany). After incubation of 100 µg/mL of total protein extract in HT29 cells, 50 µL of culture supernatant was collected from each well and added to a new 96-well plate. Then the plate was incubated with 50 µL of assay substrate (prepared by mixing 1.11 µL of catalyst for each 50 µL of Dye Solution), for 10 to 30 min, at room temperature, protect from light. The absorbance was measured at 490 nm, with 620 nm reference wavelength using a Model 96 microplate reader (GLOMAX by Promega).

Cell viability was evaluated with CellTiter 96® Aqueous Non-Radioactive Cell Proliferation Assay (Promega), using the inner salt (MTS). After removed the incubation media that will be used for LDH assay, the remaining incubation media was replaced by 120 µL MTS/PMS mix (100 µL of culture media, 19 µL of MTS and 1 µL of PMS). Then the cells were incubated at 37 °C for 1 h. Changes in absorbance were measured at a wavelength of 490 nm, using a Model 96 microplate reader (GLOMAX by Promega).

#### 4.3.3. Inhibitory Activities on HT29 Cells

##### 4.3.3.1. Determination of MMP-9 Inhibitory Activities

The fluorogenic substrate dye-quenched (DQ)-gelatin purchased from Invitrogen (Carlsbad, CA, USA) was used to quantify MMP-9 activity in the presence of total protein extracts from four species leaves. DQ gelatin was dissolved in water at 1 mg·mL^−1^ as per the manufacturer’s instructions. All solutions and dilutions were prepared in assay-buffer (50 mM Tris-HCl pH 7.6, 150 mM NaCl, 5 mM CaCl_2_ and 0.01% *v/v* Tween 20). A 96-well micro-assay plate (chimney, 96-well, black) was used. Each well was loaded with 0.1 mM (for a final volume of 200 μL) of MMP-9 (Sigma), to which were added 100 μg·mL^−1^ of total protein extracts (for a final volume of 200 μL), and the plate was incubated for 1 h at 37 °C. Subsequently, DQ-gelatin (at a final concentration of 2.5 μg·mL^−1^) was added to each well and the plate was allowed to incubate again, for 1 h. Fluorescence levels were measured (ex. 485 nm/em. 530 nm). In each experiment, both positive (without protein) and negative (without enzyme) controls were included for all samples, to correct for possible proteolytic activities present in the protein samples. All data were corrected by subtracting of their corresponding negative controls.

##### 4.3.3.2. Migration Assay on HT29 Cells

For cell migration analysis, the wound healing assay was performed. HT29 cells (5 × 10^5^ cells/well) were seeded in 6-well plates and allowed to reach to 80% confluence. After removing the media, the centers of the cell monolayers were scraped with a sterile micropipette tip to create a denuded zone (gap) of constant width. Subsequently, cellular debris was washed twice with PBS. Each well was then filled with fresh media containing the protein fractions under study, in a concentration of 50 and 100 μg protein mL^−1^ (total volume of 3 mL per well), and allowed to grow for 48 h. The wound closure was monitored and photographed at 0 and 48 h. To quantify the migrated cells, pictures of the initial wounded monolayers were compared with corresponding pictures of cells at the end of incubation. Artificial lines were drawn on pictures of the original wounds and overlaid on the pictures after 48 h incubation. Migrating cells across the lines were counted in three random fields from each triplicate treatment, and data are presented as the mean ± SD.

##### 4.3.3.3. Gelatinolytic Activity Quantification in HT29 Cell Media

After treatment with each sample as described in Section 4.3.3.2, extracellular HT29 media was collected to quantify their respective gelatinolytic activities. The assay was conducted as described in Section 4.3.3.1, but with the following alterations: each well of 96-well micro-assay plate (chimney, 96-well, black) was loaded with HT29 media from each treatment (150 μL) and the DQ-gelatin was added at a final concentration of 2.5 μg/mL (to a final volume of 200 μL). Samples were incubated at 37°C for 1 h and fluorescence levels were measured as described above (Section 4.3.3.1).

##### 4.3.3.4. Gelatin Zymography of HT29 Cell Media

To determine the metalloproteinase activities in HT29 cancer cell culture supernatants after exposure to the protein extracts, a gelatin-zymography was performed according to standard methods [37], with the following modifications: SDS-polyacrylamide gels (12.5% *w*/*v* acrylamide) were copolymerized with 1% (*w*/*v*) gelatin and the cell culture supernatants treated with a non-reducing buffer containing 62.6 mM Tris-HCl pH 6.8, 2% (*w*/*v*) SDS, 10% (*v/v*) glycerol and 0.01% (*w*/*v*) bromophenol blue were loaded into each well. Electrophoresis was carried out according to the method described by Laemmli (1970) [38] in a 12% *w*/*v* acrylamide resolving gel and a 4% *w*/*v* acrylamide stacking gel and performed in a vertical electrophoresis unit at 100 V and 20 mA per gel. After electrophoresis, gels were washed three times in 2.5% (*v/v*) Triton X-100 for 90 min each, to remove the SDS. Gels were then incubated overnight with developing buffer (50 mM Tris-HCl pH 7.4; 5 mM CaCl_2_; 1 µM ZnCl_2_ and 0.01% *w*/*v* sodium azyde), stained with Coomassie Brilliant Blue G-250 0.5% (*w*/*v*) in 50% (*v/v*) methanol and 10% (*v*/*v*) acetic acid, for 30 min and distained with a solution of 50% (*v*/*v*) methanol, 10% (*v*/*v*) acetic acid in water. White bands visible against a blue background marked the gelatinase activity of each proteinase [37].

##### 4.3.3.5. Gelatinolytic Activity Quantification in HT29 Cell Media

After treatment with each sample as described in Section 4.3.3.2, extracellular HT29 media was collected to quantify their respective gelatinolytic activities. The assay was conducted as described in Section 4.3.3.1, but with the following alterations: each well of 96-well micro-assay plate (chimney, 96-well, black) was loaded with HT29 media from each treatment (150 μL) and the DQ-gelatin was added at a final concentration of 2.5 μg/mL (to a final volume of 200 μL). Samples were incubated at 37 °C for 1 h and fluorescence levels were measured as described above (Section 4.3.3.1).

### 4.4. Polypeptide and Lectin Characterization of Leaf Protein Extracts

#### 4.4.1. Polypeptide Profile

Polypeptide profiles were characterized by SDS–PAGE (under reducing and non-reducing conditions) carried out essentially as described by Laemmli (1970) [38], in a 17.5% (*w*/*v*) acrylamide slab gels. Gels were stained with silver nitrate [39].

#### 4.4.2. Glycoprotein Detection

The glycosidic character detection the polypeptides constituents of the four extracts under study was carried out after electrophoretic run (SDS-PAGE R), followed by its transfer and immobilization in nitrocellulose membrane, by the concanavalin A-Peroxidase method, proposed by Faye & Chrispeels, in 1985 [40].

#### 4.4.3. Haemagglutination Assays with Rabbit and Human Erythrocytes

##### 4.4.3.1. Erythrocyte cells

Rabbit erythrocytes: Erythrocytes from rabbit blood were treated according to Ribeiro and colleagues (2012) [41]. Five mL of blood were washed three times in saline and incubated with trypsin for 1 h at 37 °C at a final concentration of 1% (*v*/*v*). Finally, a 4% (*v*/*v*) suspension of the trypsinised erythrocytes was stored at 4 °C and used for the haemagglutination activity assays.

Human System ABO erythrocytes: The human blood samples of ARH+, ARH-, BRH- and 0RH+ groups, were treated in a similar manner as rabbit erythrocyte. The exception is concerning with the trypsinization step that is not necessary for human erythrocytes. The final erythrocitaire solution at 4% (*v*/*v*) were preserved at 4 °C until haemagglutination assays.

##### 4.4.3.2. Measurements of Haemagglutination Activity and Haemagglutination Inhibition

For haemagglutination activity measurements [41], protein extracts (100–250 µg in 50–70 µL saline containing 2 mM CaCl_2_ and 2 mM MgCl_2_) were serially diluted (1:2) in a 96 well microplate. The erythrocyte suspension (50–70 µL) was then added and the microplate incubated for 30 min at 37 °C before visual analysis. Positive (Con-A lectin at 0.5 mg/mL) and negative (saline) controls were prepared. One haemagglutination Unit (H.U.) was defined as the minimal protein concentration which induces haemagglutination activity.

For sugar-induced inhibition of the haemagglutination activity, 50 µL to 70 µL of each sugar solution (0.1 M in saline containing 2 mM CaCl_2_ and 2 mM MgCl_2_ (except for L-fucose which was tested at 0.3 M)) was serially diluted (1:2) in a 96 well microplate. A total of 17 sugars were assayed: (1) D-glucose, (2) D-glucosamine, (3) *N*-acetyl-D-glucosamine; (4) D-galactose; (5) D-galactosamine; (6) *N*-acetyl-D-galactosamine (7) lactose; (8) D-mannose; (9) raffinose; (10) L-fucose (0.3 M); (11) melezitose; (12) α-methyl-D-glucopyranoside; (13) α-methyl-D-mannopyranoside (14) sacarose; (15) maltose (16) sialic acid and (17) L(-)-fucose (0.3 M). After sugar dilution was added 4 H.U. of the protein under study (in 50 to 70 µL saline containing 2 mM CaCl_2_ and 2 mM MgCl_2_), and the plate was sealed and incubated for 1 h at room temperature. Finally, the erythrocyte suspension at 4% (*v*/*v*) (in 50 to 70 µL saline containing 2 mM CaCl_2_ and 2 mM MgCl_2_ was added and the microplate incubated for 1 h at room temperature. The agglutination of the erythrocytes was macroscopically visible and recorded. Appropriate controls were used as described before. Sugar specificity was evaluated by the minimum sugar concentration capable of inhibiting the haemagglutination activity (m.i.c).

### 4.5. Lectin Binding to HT29 Cell Glycosylated Receptors

#### 4.5.1. Isolation of HT29 Cell Membranes

HT29 cells were grown as described previously, and the membranes were isolated by the method described by Edouart and colleagues (2008) [42]. The cells preserved at −80 °C (26 × 10^6^ cells) were thawed rapidly at 37 °C, and the suspension of cells was washed with 10 volumes of HES buffer (20mM Hepes, pH 7.4, 250mM Sacarose), by centrifugation at 750× *g* for 10 min, at 21°C (Beckman J2-21 m/E, Rotor JA 20000, (Indianopolis, USA). The supernatant was discarded and the pellet was washed twice by ressuspending HES buffer with the addition of a protease inhibitor cocktail (without EDTA, Roche). Cell lysis was achieved by cryolise, in which cells were subject to series of freezing and thawing cycles (4 times), during 30min, at −20 °C, associated with sonication in ultrasound during 20 min, followed a centrifugation at 960× *g* for 10 min at 4 °C, and the sediment was discarded. The final supernatant was ultracentrifuged at 100,000× *g* for 45 min, 4 °C (Beckman J2-21 m/E, Rotor JA 20000). The sediment containing the cell membranes, was solubilized in 2 mL of physiological saline (0.9% NaCl), containing 2mM of CaCl_2_ and 2mM of MgCl_2_ and was divided into aliquots containing 1 mg protein determined by the Bradford method (1976) and kept at −80 °C, until use.

#### 4.5.2. Affinity Binding of Polypeptides/Lectins to HT29 Cells Membranes

The isolated HT29 cells membranes were incubated individually with the total protein leaf extracts. One mg of HT29 protein membrane was solubilized in 3.5 mL of saline containing 2 mM CaCl_2_ and 2 mM MgCl_2_ and incubated during 35 min at 25 °C, by gentle agitation, with 1.4 mg of total proteins leaves extracts dissolved in 4 mL of saline containing 2 mM CaCl_2_ and 2 mM MgCl_2_. After incubation, in order to remove the proteins unbound to membranes, one prior centrifugation at 12,100× *g* was performed for 10 min, followed by a cycle of three consecutive washes of the sediment, with 15 volumes of saline containing salts, by centrifugation at 12,000× *g* for 10 min at 4 °C (Beckman J2-21 m/E, Rotor JA 20000). The supernatant was discarded and the obtained pellet, consisting of the lectins bound to membranes, was subsequently solubilized in saline solution (containing salts) and used, after protein measurement, to detect the lectins bounded to membranes. The control was made by using saline instead total protein extracts in the membrane incubation.

#### 4.5.3. Evaluation of Lectin Binding to HT29 Cell Membranes by 2D Analysis

Two-dimensional (2D) electrophoresis (IEF/SDS-PAGE) of controls and cell membranes incubated with *Juniperus oxycedrus* subsp. *oxycedrus*, *Juniperus oxycedrus* subsp. *badia* and *Arbutus unedo* protein extracts were carried out as follows:

In the first dimension, isoelectric focusing (IEF), 1 g of protein incubation product was separated using the IPGphor system (Amersham Pharmacia). ImmobilineDrystrip gel strips (IPG strips) (13 cm, pH 3–10) were obtained from Amersham Pharmacia. IPG strips were rehydrated with 250 µL of a solution containing 0.5% (*v*/*v*) IPG-buffer pH 3–10, 7 M urea, 2 M thiourea, 2% (*v*/*v*) NP-40, 1% (*v*/*v*) dithiothreitol and protein sample in the IPGphor strip holders. The program used for IEF included the following steps: rehydration 30 V·h, 12 h; step 1 - 250 V·h, 1 h; step 2 - 500 V h, 2 h; step 3 - 1000 V·h, 2 h; step 4 - 2500 V·h, 3.5 h; step 5 (gradient) – 8000 V·h, 1 h; and step 6 – 8000 V during 25 min. After focusing, the gel strips were immediately frozen at −80 °C.

The second dimension (R-SDS-PAGE) was performed as described above except that the gel contained only the separating gel. After IEF, the gel strips were thawed and equilibrated for 15 min, with agitation, in 50 mM Tris-HCl buffer, pH 8.8, containing 6 M urea, 30% (*v*/*v*) glycerol, 2% (*w*/*v*) SDS and 1% (*w*/*v*) dithiothreitol. The strips were subsequently equilibrated for another 15 min, with agitation, in a similar solution that contained 2.5% (*w*/*v*) iodoacetamide instead of dithiothreitol, placed on top of a 17.5% (*w*/*v*) acrylamide SDS-PAGE gel, sealed with 0.7% (*w*/*v*) agarose (containing 0.002% (*w*/*v*) bromophenol blue) and electrophoresed (220 V, 15 mA for 15 min followed by 220 V, 30 mA). After electrophoresis, the gels were stained by silver nitrate [39]. A comparative study was made between 2D HT29 cell membrane control and 2D of cell membranes incubation with the leaves extracts, in order to evaluate the polypeptide(s) bounded to the membranes.

### 4.6. Protein Quantification

Protein concentrations were measured by Bradford method [43], with bovine serum albumin used as the standard.

### 4.7. Statistical Analysis

All experiments were performed in triplicate, in at least three independent times, and the data were expressed as the mean ± standard deviation (SD). SigmaPlot software (version 12.5) was used for comparing different treatments, using one-way and two-way analysis of variance (ANOVA). Tukey’s test was used to compare differences between groups and the statistical differences with *P* value < 0.05 where considered significant.

## 5. Conclusions

Overall, results presented here provide evidence that the protein extracts of all species under study present a lectin-like inhibitory action on cancer cell invasion, MMP-2 and MMP-9 activity in HT29 cancer cells, as well as the capacity to induce cell death and decrease cell viability, which can be potentially used as an anticancer-agent. The lectins binding to the HT29 cells membrane receptors of the different extracts led us to hypothesize that the HT29 cellular death was due to a mechanism related to lectins. The results presented here provide evidence that the leaf protein extracts of *Juniperus oxycedrus* subsp. *badia* and *Arbutus unedo* present a lectin-like inhibitory action on cancer cell migration and MMP-2 and MMP-9 activity in HT29 cancer cells, which is unique to this subspecies, and which can be potentially used as an anticancer-agent. In particular, the specific binding of a 30, 32 and 50 kDa polypeptides from this species to glycosylated receptors of HT29 cell membranes indicates lectin activity, resulting in an antitumor effect by MMPs inhibition. The presence of a 50 KDa specific polypeptide with lectin activity that mark the difference between Joox and Joba activities seems to be the clue for further research. These results open novel possibilities for the use of *Juniperus* species protein extracts and *Arbutus unedo* as anticancer agents in both prevention as well as targeted-therapy in colon cancer.

As glycan-binding proteins can deliver intracellular signals or control extracellular processes that promote initiation, execution and resolution of cell death programs, the evolution of these results with in vivo assays in animal model of sever colitis and colon carcinoma open novel possibilities for the use of these species protein extracts as anticancer agents in both prevention as well as targeted-therapy in colon cancer.

## Figures and Tables

**Figure 1 ijms-20-03059-f001:**
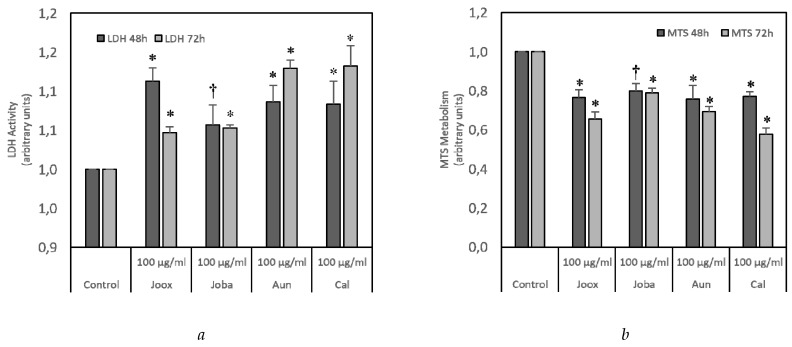
Screening of antitumor activity. HT29 cells were exposed to 100 µg/mL of total protein extract from *Juniperus oxycedrus* subsp. *oxycedrus* (Joox), *Juniperus oxycedrus* subsp. *badia* (Joba), *Arbutus unedo* (Aun) and *Corema album* (Cal), for 48 and 72 h. Saline containing 2 mM CaCl_2_ and 2 mM MgCl_2_ was used as control. General cell death (**a**) and cell viability (**b**) were evaluated by LDH release and MTS metabolism assays, respectively. Results are expressed as mean ± SEM fold-change to control from three independent experiments. * *p* < 0.01 and ^†^ <0.05 for HT29 cells.

**Figure 2 ijms-20-03059-f002:**
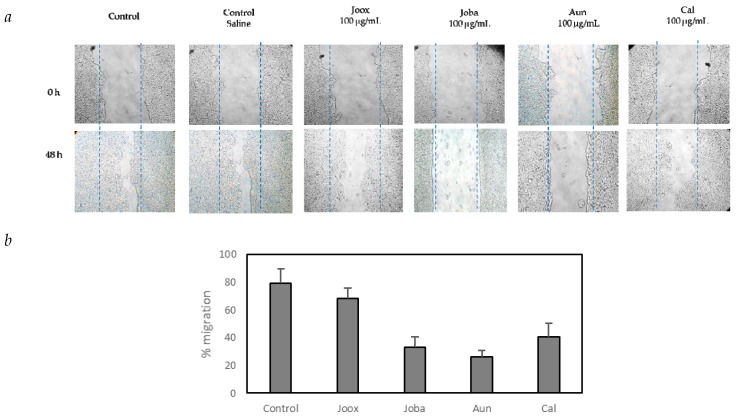
HT29 cell migration after exposure to total soluble protein extracts of *Juniperus oxycedrus* subsp. *oxycedrus* (Joox), *Juniperus oxycedrus* subsp. *badia* (Joba), *Arbutus unedo* (Aun) and *Corema album* (Cal), as determined by migration assays. Cells were grown until reaching 80% confluence and the monolayer was scratched with a pipette tip (day 0). Cells were then exposed to 100 µg protein mL^−1^ protein extract and cell migration was recorded after 48 h (**a**). The histogram reports the relative migration rates, where values are the means of at least three replicate experiments ±SD and are expressed as % wound closure in relation to day 0 (**b**).

**Figure 3 ijms-20-03059-f003:**
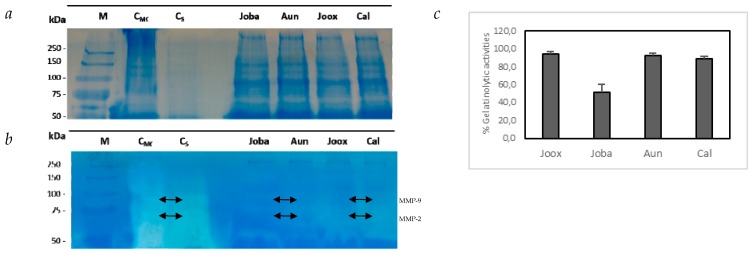
Gelatinolytic activity for all total protein extracts carried was out on SDS-PAGE NR gel. (**a**) Control gel on polyacrylamide of 12.5% (*m*/*v*), staining by CBB G250, representative of samples from the extracellular medium after incubation with *Juniperus oxycedrus* subsp. *badia* (Joba), *Arbutus unedo* (Aun), *Juniperus oxycedrus* subsp. *oxycedrus* (Joox) and *Corema album* (Cal) extracts: 5 μL from the molecular weight marker (M) and 20 μL extracellular medium were applied of each sample as for medium controls, complete medium control (CMC) and control with saline (CS). (**b**) Zymographic test carried out on 12.5% (*m*/*v*) polyacrylamide gel containing 1% (*m*/*v*) gelatin and staining by CBB G250, representative of samples from the extracellular medium, collected at the end of the inhibition test of cell migration. 5 μL of molecular weight marker (M) and 20 μL of all samples were also applied. (**c**) Proteolytic activity of gelatinases present in the HT29 extracellular media after a 48 h-exposure to 100 μg·mL^−1^ of buffer-soluble fractions from all species, as quantified by the DQ fluorogenic method. Results are expressed as relative fluorescence as a % of controls and represent an average of at least three replicate experiments ± SD (**c**). The arrows highlighted the presence of metalloproteinases MMP-9 and MMP-2.

**Figure 4 ijms-20-03059-f004:**
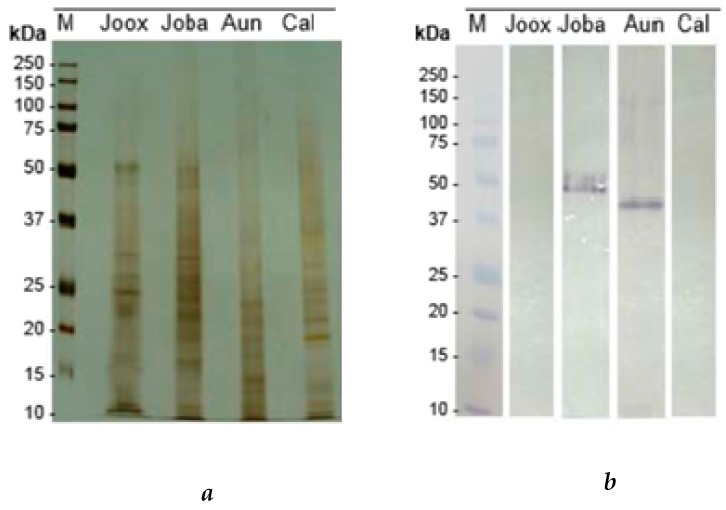
Glycoprotein detection in protein leaves extracts. Polypeptide profile of *Juniperus oxycedrus* subsp. *oxycedrus* (Joox), *Juniperus oxycedrus* subsp. *badia* (Joba), *Arbutus unedo* (Aun) and *Corema album* (Cal) leaves extracts in a 17.5% SDS-PAGE-R (*m*/*v*) acrylamide, stained by AgNO3 as gel control. It was applied 7µg of each extract and 3µg of molecular marker (a). Transfer and glycodetection in a nitrocellulose membrane of leaves extracts: were transferred 540 μg of Joox, 700 μg of Joba, 350 μg Aun and 400 μg of Cal, of the respective extracts (b).

**Figure 5 ijms-20-03059-f005:**
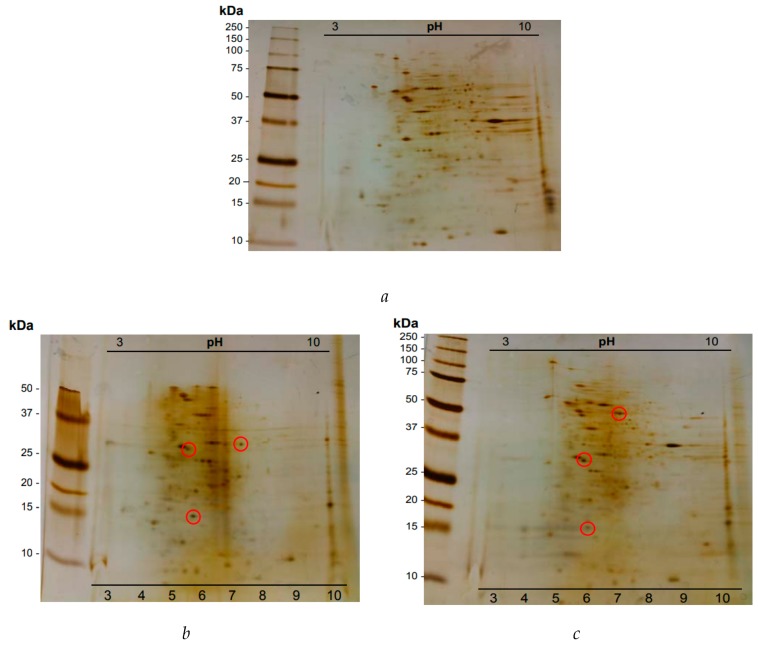
Two-dimensional analysis (IEF/SDS-PAGE R) of lectin-bound proteins of HT29 cell membrane. (**a**) Two-dimensional electrophoresis of the HT29 cell membrane control, 1 mg of membrane protein was applied. (**b**)Two-dimensional electrophoresis of the total protein extract of *Juniperus oxycedrus* subsp. *Oxycedrus*, (**c**), *Juniperus oxycedrus* subsp. *badia*, (**d**) *Arbutus unedo* and (**e**) *Corema album*, obtained after incubation with 1 mg of HT29 cell membrane with 1.6 mg protein extract and applied to a 17.5% (*w*/*v*) gel of acrylamide, stained by AgNO_3_.

**Table 1 ijms-20-03059-t001:** Haemagglutination activity detection in human and rabbit erythrocytes.

Plant Specie	Blood Source (4%)	A		B		0		Haemagglutination Activity (H.U.) (µg)
		RH+	RH-	RH+	RH-	RH+	RH-	
*Juniperus oxycedrus* subsp. *oxycedrus*	Human	(+)	ND	ND	(-)	(-)	ND	ARH + = 16.6
		ND	(+)	ND	ND	ND	ND	ARH - = 250
	Rabbit	------------------------------------------------------------------						ND
*Juniperus oxycedrus* subsp. *badia*	Human	(+)	ND	ND	(-)	(-)	ND	ARH + = 50
		ND	(+)	ND	ND	ND	ND	ARH - = 27.7
	Rabbit	------------------------------------------------------------------						0.205
*Arbutus unedo*	Human	(-)	ND	ND	(-)	(-)	ND	ND
		ND	(-)	ND	ND	ND	ND	ND
	Rabbit	------------------------------------------------------------------						0.045
*Corema album*	Human	(-)	ND	ND	(-)	(-)	ND	ND
		ND	(-)	ND	ND	ND	ND	ND
	Rabbit	------------------------------------------------------------------						0.069

(+) exhibited haemagglutination activity; (-) did not exhibit haemagglutination activity; ND assay not performed. Note: Blood group AB was not tested.

**Table 2 ijms-20-03059-t002:** Haemagglutination activity inhibition by sugars.

Sugars		*Juniperus Oxycedrus* subsp. *badia* 4 H.U.	*Arbutus unedo* 4 H.U.	*Corema album* 4 H.U.
No.	(0.1 M)	Sugar Minimal Inhibitory Concentration (m.i.c.) (M)		
1	D-Glucose	0.033	0.074	0.1
2	D-Glucosamine	3.7 × 10^−3^	8.2 × 10^−3^	UD
3	*N*-Acetyl-D-glucosamine	11.1 × 10^−3^	0.025	0.1
4	D-Galactose	UD	UD	UD
5	D-Galactosamine	3.7x10^−3^	0.025	0.1
6	*N*-Acetyl-D-galactosamine	UD	UD	UD
7	Lactose	UD	UD	UD
8	D-Mannose	5.6 × 10^−7^	0.074	3.7 × 10^–3^
9	Raffinose	0.4 × 10^−3^	UD	UD
10	L-Fucose*	0.1 × 10^−3^	UD	UD
11	Melezitose	5.6 × 10^−7^	2.7 × 10^−3^	1.5 × 10^–5^
12	α-Methyl-D-glucopyranoside	5.6 × 10^−7^	0.025	0.4 × 10^–3^
13	α-Methyl-D-mannopyranoside	5.1 × 10^−7^	2.7 × 10^−3^	1.5 × 10^–5^
14	Sucrose	4.6 × 10^−6^	0.1	0.011
15	Maltose	UD	0.074	ND
16	Sialic acid	UD	UD	UD
17	L(-)-Fucose *	4.0 × 10^−4^	0.011	ND

ND assay not performed; UD undetectable. * Were applied 0.3 M of fucose.

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
