# Peer review of "New Lectins from Mediterranean Flora. Activity against HT29 Colon Cancer Cells"

_ijms, 2019, doi:10.3390/ijms20123059_

Round 1
Reviewer 1 Report
In this manuscript, Oliveira et al. selected lectins exhibiting antitumoral activity from Mediterranean plant species and addressed their possible mechanisms in HT29 colon cancer cells. Overall the manuscript is quite well designed and provides impressive results demonstrating that lectins from J. oxycedrus badia and Arbutus unedo extracts exert the inhibitory activity of gelatinase.
Major points
(1) Figure 2. The wound healing assay is more appropriately termed a “scratch assay” or "migration assay" as "wound healing assay" tend to be in vivo. Moreover, this experiment is not an invasion assay but just a migration assay. So, it is best to avoid the term "invasion". The authors should check the entire manuscript and replace the "invasion" with the appropriate words.
(3) Figure 3b. Gelatin zymography is not convincing. This reviewer is troubled by the apparent lack of specific white bands on the background. More clear zymographic images should be provided.
Author Response
(1) Figure 2. The wound healing assay is more appropriately termed a “scratch assay” or "migration assay" as "wound healing assay" tend to be in vivo. Moreover, this experiment is not an invasion assay but just a migration assay. So, it is best to avoid the term "invasion". The authors should check the entire manuscript and replace the "invasion" with the appropriate words.
Answer:
All manuscript are checked and all terms replaced by migration assay- Done
(3) Figure 3b. Gelatin zymography is not convincing. This reviewer is troubled by the apparent lack of specific white bands on the background. More clear zymographic images should be provided.
Answer:The change of image with new appointment of the position of MMPs, in order to clarify the results-Done
Reviewer 2 Report
It is an interesting paper well designed and the methodology properly carried out, including an ample variety of techniques. the authors have addressed the in vitro bioactivities of leaf protein extracts of four plants as well as their potential lectin activity in order to test their possible use in targeted antitumor therapy, based on their effects on culture of colon cancer cells HT29.
The values of standard deviations should appear in the text beside the means and the authors must explain why they have used a parametric statistical analysis like ANOVA, taking into account the scarce number of measurements performed for each variable.
The discussion is too long because the authors repeat what is said in the results.
The conclusions are original and consistent with the results and stimulate further research on these lectins to evaluate the extent of their putative use as anticancer agents.
The English style should be revised and a number of misspellings corrected.
The word "specie" must be replaced by "species" throughout the text.
line 45: act instead of acts
line 63: one "and" should be deleted
line 190: "three" should be deleted
line 192: exhibited instead of exhibit
line 198: revealed instead of reveal
line 212: leaf instead of leave
line 215: sentence is incomplete
line 270: "Were" must be deleted
line 309: A subject is needed in the sentence
line 316: to instead of for
line 317: reveals instead of reveal
lines 320, 359, 374, 375, 379: other instead of others
line 361: important instead of importance
lines 387, 431: present instead of presents
line 464: leaf instead of leaves
line 677: animal instead of animals.
Author Response
Point 1-The values of standard deviations should appear in the text beside the means and the authors must explain why they have used a parametric statistical analysis like ANOVA, taking into account the scarce number of measurements performed for each variable.
Response:We did the statistical analysis by ANOVA to use greater rigor in approach and make the results more robust
Point 2- The discussion is too long because the authors repeat what is said in the results.
Response:We are in agreement. The discussion was remade and shortened.-Done
Point 3- The English style should be revised and a number of misspellings corrected.
Response: Done
Point 4- List of terms to correct.
Response: All of them were cheked and corrected-Done